# Toxicological Evaluation of Kaempferol and Linearolactone as Treatments for Amoebic Liver Abscess Development in *Mesocricetus auratus*

**DOI:** 10.3390/ijms251910633

**Published:** 2024-10-02

**Authors:** Luis Varela-Rodríguez, Fernando Calzada, José Antonio Velázquez-Domínguez, Verónica Ivonne Hernández-Ramírez, Hugo Varela-Rodríguez, Elihú Bautista, Mayra Herrera-Martínez, Diana Laura Pichardo-Hernández, Rodrigo Daniel Castellanos-Mijangos, Bibiana Chávez-Munguía, Patricia Talamás-Rohana

**Affiliations:** 1Facultad de Ciencias Químicas (FCQ), Universidad Autónoma de Chihuahua (UACH), Chihuahua 31125, CP, Mexico; lvrodriguez@uach.mx (L.V.-R.); hvrodriguez@uach.mx (H.V.-R.); 2Unidad de Investigación Médica en Farmacología, Hospital de Especialidades UMAE-CMNSXXI-IMSS, Ciudad de México 06725, CP, Mexico; fercalber10@gmail.com; 3Departamento de Infectómica y Patogénesis Molecular, CINVESTAV-IPN, Ciudad de México 07360, CP, Mexico; jauam14@yahoo.com.mx (J.A.V.-D.); vhernandezr@cinvestav.mx (V.I.H.-R.); laurita_nesc@hotmail.com (D.L.P.-H.); bchavez@cinvestav.mx (B.C.-M.); 4Facultad de Medicina y Ciencias Biomédicas, Universidad Autónoma de Chihuahua (UACH), Chihuahua 31109, CP, Mexico; 5Unidad de Ciencias Ambientales, IPICYT, San Luis Potosí 78216, CP, Mexico; hormigaqfb@hotmail.com; 6Instituto de Farmacobiología, Universidad de la Cañada (UNCA), Teotitlán de Flores Magón, Oax 68540, CP, Mexico; chimay_2002@hotmail.com; 7Unidad de Imagenología Diagnóstica, Centro Médico ISSEMyM-Arturo Montiel Rojas, Metepec 52170, CP, Mexico; rodan_casmij@hotmail.com

**Keywords:** amoebic liver abscess, *Entamoeba histolytica*, kaempferol, linearolactone, *Mesocricetus auratus*, toxicity

## Abstract

Several studies with kaempferol (KP) and linearolactone (LL) have demonstrated their antiparasitic activity. However, the toxicity of these treatments is unknown. Therefore, this study aimed to evaluate the possible toxicological effects of intraperitoneal (i.p.) administration of KP or LL on the amoebic liver abscess model (ALA) in *Mesocricetus auratus*. An ALA was induced in male hamsters with 1.5 × 10^5^
*Entamoeba histolytica* (*E. histolytica*) trophozoites inoculated in the left hepatic lobe. The lesion evolved for 4 days, and then KP (5 mg/kg body weight/day) or LL (10 mg/kg body weight/day) was administered for 4 consecutive days. Then, magnetic resonance imaging (MRI), paraclinical analyses, and necropsy for histopathological evaluation were performed. There was similar ALA inhibition by KP (19.42%), LL (28.16%), and metronidazole, the antiamoebic control (20.87%) (*p* ≤ 0.05, analysis of variance [ANOVA]). There were hepatic and renal biochemical alterations in all treatment groups, mainly for KP (aspartate aminotransferase: 347.5 ± 37.5 U/L; blood urea nitrogen: 19.4 ± 1.9 g/dL; *p* ≤ 0.05, ANOVA). Lesions found in the organs were directly linked to the pathology. In conclusion, KP and LL decreased ALA development and exerted fewer toxicological effects compared with metronidazole. Therefore, both compounds exhibit therapeutic potential as an alternative treatment of amoebiasis caused by *E. histolytica*. However, additional clinical studies in different contexts are required to reaffirm this assertion.

## 1. Introduction

Diarrheal diseases are a significant public health issue throughout the world due to the high morbidity and mortality rates among children [1,2]. The etiological agents of diarrhea include bacteria (e.g., *Escherichia coli*), viruses (e.g., *Rotavirus*), and protozoa (e.g., *Giardia lamblia* and *Entamoeba histolytica*) [1]. The latter pathogen causes acute diarrhea, dysentery, fulminant colitis, amebomas, and liver abscesses [2].

The pathogenesis of amebiasis starts with the infection and subsequent spread of the protozoan through the intestine and extraintestinal areas [1]. *E. histolytica* crosses the intestinal epithelium and migrates to the liver, leading to the development of an amoebic liver abscess (ALA) [2], a common complication of amebiasis. An ALA is characterized by a purulent infiltrate in various liver areas, which triggers a significant inflammatory response, characterized by the presence of polymorphonuclear leukocytes (PMNs), mainly neutrophils in the acute phase, and mononuclear cells such as macrophages in the chronic phase [2]. Clinically, the most common treatment is metronidazole (MTZ). However, several in vitro and in vivo studies have emphasized the development of resistance mechanisms to MTZ [3,4]. Additionally, MTZ has been associated with several medical complications, including hypersensitivity, neutropenia, peripheral neuropathies, and cancer [5,6]. This has led to the search for new antiparasitic drugs that do not compromise the morphophysiological integrity of body compartments or present high toxicity [7].

In this context, traditional medicine offers a promising avenue to discover new effective therapeutic agents [8]. Our research group has recently reported that kaempferol (KP, 3,5,7-trihydroxy-2-(4-hydroxyphenyl-4H-1-benzopyran-4-one) and linearolactone (LL, (4S,7R,9R,10S)-7-(furan-3-yl)-9-methyl-6,16-dioxatetracyclo[8.7.0.01,14.04,9]heptadeca-11,13-diene-5,15-dione) inhibit the formation of an ALA [9]. KP is a flavonoid that effectively inhibits *E. histolytica* trophozoites (half-maximal inhibitory concentration [IC_50_]: 27.7 µM, 48 h) [10]. The primary mechanism of action of KP involves altering vital cellular functions in the parasite, such as adhesion, migration, and phagocytic activity [10,11]. This is achieved through the overexpression of cytoskeletal proteins such as actin and myosin-II, as well as other relevant proteins including cortexillin-II, a heat shock protein, and glyceraldehyde phosphate dehydrogenase and fructose-1,6-bisphosphate aldolase, which are related to the parasite metabolism [11]. On the other hand, LL is a *neo*-clerodane-type diterpene that has been shown to exert antiparasitic activity [12]. LL has demonstrated greater antiproliferative activity against *E. histolytica* trophozoites (IC_50_: 22.9 µM, 48 h) [12,13]. However, the exact mechanism of action of this compound has not yet been fully elucidated. Studies have indicated that LL induces cellular alterations, such as chromatin condensation, intracellular reactive oxygen species (ROS) production, and loss of cellular structures [9]. This compound has also been shown to induce apoptosis in *E. histolytica* [12,13].

To date, there are no studies that have investigated the potential toxicological effects of KP and LL during the treatment of amoebiasis in preclinical or clinical assays. Toxicological evaluations of antiamoebic therapies are relevant because *E. histolytica* can induce an ALA [2]. ALA development limits several vital liver functions, including metabolism, immunity, digestion, detoxification, and vitamin storage [14]. Therefore, severe liver dysfunction in patients with an ALA can lead to various medical complications, such as hepatic encephalopathy, jaundice, toxicity, infections, multiple organ failure, and, ultimately, death [15]. In this regard, the present study has specifically emphasized bioinformatics predictions concerning the molecular targets for KP and LL, as well as evidence on the selectivity, toxicity, and histopathological responses toward both compounds in animal models of ALA.

This study used the Syrian hamster (*Mesocricetus auratus*) as an animal model, considering that numerous studies support its efficacy in replicating the onset, progression, and symptoms of various human diseases [16]. Specifically, research indicates that direct intrahepatic inoculation of *E. histolytica* trophozoites enables the development of an ALA similar to what is observed in humans with extraintestinal amoebiasis [16]. Thus, animal models such as the Syrian hamster represent valuable tools for pharmacological research aimed at developing new therapeutic candidates for this disease [16]. The aim of the present study was to evaluate the possible toxicological effects of the intraperitoneal (i.p.) administration of KP and LL in *M. auratus* during ALA development.

## 2. Results

### 2.1. Cytotoxic Effect of the Active Principles KP and LL in Cell Lines

First, the effect of KP and LL on the viability of *E. histolytica* as well as the CHO-K1 and BEAS-2B cell lines was assessed. These two normal cell lines were selected because they are widely used in toxicological assays [17] and have a different cellular origin (mammalian and human epithelium), which allows the cytotoxicity of KP and LL to be evaluated in a broader context. Flow cytometry showed that 31.6 and 7.8 µg/mL of KP and LL, respectively, reduced the viability of virulent HM-1 strain by 50% compared with MTZ (positive control) or 1× phosphate-buffered saline (PBS, the negative control and vehicle [Vh]) after treatment for 48 h (*p* ≤ 0.05, ANOVA followed by the Dunnett’s test) (Table 1 and Appendix A). Regarding normal cells, KP had a stronger inhibitory effect on the CHO-K1 and BEAS-2B cells (IC_50_: 70 and 100 µg/mL, respectively) compared with LL (IC_50_: 300 and 200 µg/mL, respectively) and Vh after treatment for 24 h (*p* ≤ 0.05, ANOVA followed by the Dunnett’s test) (Figure 1). Finally, KP and LL showed a theoretical median lethal dose (LD_50_) of 552 and 824 mg/kg body weight, respectively, for the oral route, while MTZ presented an LD_50_ of 1026 mg/kg body weight (Table 1).

### 2.2. Treatment with KP or LL Selectively Induced Cell Death and Morphological Alterations in Trophozoites but Not in Normal Cells

Flow cytometry showed that treating *E. histolytica* trophozoites with MTZ (0.16 μg/mL), KP (31.6 μg/mL), or LL (15.6 μg/mL) decreased the size and granularity of the cell population by 52%, 55%, and 66%, respectively (Appendix A), compared with the Vh group after 48 h in a dose-dependent manner (*p* ≤ 0.05, ANOVA followed by the Dunnett’s test). Under the same treatment conditions, some trophozoites were apoptotic or necrotic when treated with KP (8.2% and 45.2%, respectively) or LL (4.3% and 90.1%, respectively); these changes occurred in a dose-dependent manner (Appendix A). These effects were similar with MTZ (2% and 62.2%, respectively) compared with the Vh group in both cases (*p* ≤ 0.05, ANOVA followed by the Dunnett’s test). Transmission electron microscopy (TEM) and confocal microscopy corroborated these findings, revealing the binding of annexin-V to phosphatidylserine in all cells treated with the active compounds (Appendix A, upper panel). Confocal microscopy revealed that KP or LL treatment induced mitotic arrest (binary fission type) and altered the actin cytoskeleton, leading to atypical cells (Appendix A). In particular, cells treated with KP presented disorganized actin microfilaments, while cells treated with LL presented a reduction in polymerized actin (Appendix A, middle panel). This abnormality was similar to the MTZ control, but all treatments differed from the polarized cortical actin observed in the Vh group. Finally, TEM suggested ultrastructural modifications in *E. histolytica* (Appendix A). The trophozoites treated with MTZ, KP, or LL under the same dose conditions showed a reduction in the size and number of vacuoles; rearrangement of euchromatin; an increase in glycogen granules; and the presence of membranous structures such as autophagosomes or apoptotic bodies, arranged in the nuclear and vacuolar cavity and the cytoplasm, compared with the Vh group (Appendix A, lower panel).

### 2.3. The Effect of Treatment with KP or LL in the M. auratus ALA Model

The amoebicidal effect of KP and LL against ALA has been studied in hamsters [9,10,11,12,13]. The hamsters did not present a crisis episode associated with behavioral changes (e.g., agitation, tremor, asthenia, and anorexia, among others), symptoms of intoxication (e.g., dyspnea, photophobia, blindness, diarrhea, tachycardia-arrhythmias, dystonia, myasthenia, seizures, and epithelial pigmentation, among others), or anasarca (related to congestive heart failure, liver failure, or kidney failure) 24 h after treatment. The hamsters did not show jaundice, pruritus, or chills. However, body temperature was not recorded to adequately rule out a febrile episode. The hamsters were weighed at all times during the trial, but there were no changes in body weight or significant differences between the treated and untreated groups (*p* ≥ 0.05, ANOVA followed by the Dunnett’s test) (Table 2; Appendix A). Only the LL group showed a slight decrease of −0.21% in body weight compared with the control (disease-free) group (Appendix A). Thus, the clinical conditions associated with the progression of an ALA were ruled out, including anorexia due to general malaise (fever, chills, or muscle stiffness), abdominal pain (due to distension), and cachexia (due to acute diarrhea and bleeding/mucous membranes) [18]. Likewise, the antiparasitic activity results indicated that there were no significant differences in the body weight of the ALA model and no signs of toxicity that manifested in a change in body weight. There was 100% survival after the administration of each treatment (Table 2).

### 2.4. Paraclinical Analysis of Post-Treatment Liver and Kidney Function

The effect of KP and LL was evaluated during the course of ALA development based on biochemical and hematological measures (Figure 2 and Appendix A). The findings from the KP and LL groups were compared with the Vh group (ALA without treatment), the disease-free group (without ALA), and the reference ranges reported for hamsters [19,20]. There was hypertriglyceridemia in all experimental groups with respect to the reference values (30–145 mg/dL), while the Vh and KP groups showed higher triglyceride levels compared with the disease-free group (*p* ≤ 0.05, ANOVA followed by the Tukey test) (Appendix A). Furthermore, the cholesterol levels were elevated in the Vh and KP groups (163 ± 21 and 164 ± 43 mg/dL, respectively) compared with the disease-free group (*p* ≤ 0.05, ANOVA followed by the Dunnett’s test). It is important to highlight that these values are not outside of the reference range for hamsters (94–237 mg/dL), and these data correlate with what has been reported previously [21,22]. Regardless of the treatment, the protein levels remained within the reference range (4–8.6 g/dL), and there were no alterations in the treatment groups compared with the disease-free group (Figure 2A). Finally, hepatic and renal functions were evaluated to determine the disease progression, the possible toxicity associated with each treatment, and liver tissue damage as a result thereof (Appendix A).

#### 2.4.1. Evaluation of the Hepatic Function

The liver profile included the gamma glutamyl-transpeptidase (GGT), albumin (AL), alkaline phosphatase (ALP), aspartate aminotransferase (AST), alanine aminotransferase (ALT), total bilirubin (TB), and direct bilirubin (DB) serum levels, according to recommendations of the National Academy of Clinical Biochemistry and the American Association for the Study of Liver Diseases [23]. All groups showed differences in the AL, ALP, and DB serum levels compared with the reference data, although with some exceptions in the disease-free group (*p* ≤ 0.05, ANOVA followed by Dunnett’s test) (Appendix A).

Several of the groups presented hypoalbuminemia, hyperphosphatasemia, hyperbilirubinemia, and hypoazotemia. They were most pronounced in the Vh group, followed by KP, MTZ, and LL groups (Figure 2B). The Vh and MTZ groups had more extensive liver damage compared with the other groups (*p* ≤ 0.05, ANOVA followed by the Tukey test). This liver damage was associated with ALA development for the Vh group (mainly) and the KP group (to a lesser extent), while the damage presented in the MTZ and LL groups appeared to be more directly associated with the treatments themselves, due to the correlation among the histopathological (Appendix A) and paraclinical findings (Figure 2B) (*p* ≤ 0.05, ANOVA followed by the Tukey test). These data were supported by the presence of hyperphosphatasemia and hypertransaminasemia (mainly due to ALT), with a direct correlation with hepatocellular necrosis and, therefore, an indirect correction with the ALA (Appendix A).

The groups presented lower values of ALP, AST, and ALT compared with the Vh group (*p* ≤ 0.05, ANOVA followed by the Tukey test), except for the similar AST in the LL group (163 ± 22 U/L) (*p* > 0.05, ANOVA followed by Dunnett’s test) and ALT in the MTZ group (135 ± 23 U/L, exceeding the reference range) (*p* > 0.05, ANOVA followed by Dunnett’s test) (Figure 2B). However, these liver parameters remained higher in all groups compared with the disease-free group (Figure 2B) due to active cell necrosis (chronic type) and lack of remission but with stable progression of the ALA (Appendix A). This outcome was correlated with the serum levels of some proteins synthesized in the liver, namely GGT and AL (Figure 2B).

High TB, DB, GGT, and ALP levels have been correlated with the size of an ALA [23]. Consistently, the Vh group presented the largest ALA, followed by the KP, MTZ, and LL groups (*p* ≤ 0.05, ANOVA followed by Dunnett’s test) (Figure 2B and Table 2). These results correlated with the histopathological findings (Appendix A). Moreover, transaminases, ALP, and GGT have been correlated with hepatic obstruction (macro and micro), a common complication during ALA development due to local necrosis and inflammation that can occur due to the pathology [24]. However, unlike the histopathological analysis, the paraclinical analysis could not demonstrate hepatic obstruction. Specifically, the GGT, cholesterol, and TB levels were high but within the reference range. It is possible that treatment with KP or LL induces a period of metabolic compensation that may later trigger complications associated with cholestases. However, studies with longer exposure times are necessary to evaluate this possibility.

#### 2.4.2. Evaluation of the Renal Function

The biochemical parameters of renal function in the different treatments presented a situation that was opposite to that observed for liver function (Figure 2B). The urea levels were lower for the treatment groups compared with the disease-free group (Appendix A). Furthermore, compared with the disease-free group, the Vh group presented a considerable decrease in the blood urea nitrogen (BUN) level (6.5 ± 0.5 g/dL), and the MTZ group presented a significant increase in the creatinine (CR) level (1.01 ± 0.2 mg/dL) (*p* ≤ 0.05, ANOVA followed by the Tukey test) (Figure 2B). The hypoazotemia observed in the different treatment groups may be related to the inability of the liver to metabolize dietary proteins due to liver failure caused by ALA progression [25].

Based on the data, it is possible to deduce that the Vh group presented less resolution of the ALA lesion, followed by the LL, KP, and MTZ groups. The hypercreatinemia observed in the MTZ group may be associated with early renal dysfunction or acute renal failure due to complications related to the ALA and toxicity from the treatment itself [26]. Previous studies have shown that MTZ is eliminated via the kidney and can accumulate in the body by altering glomerular filtration or tubular secretion [27]. This process can occur when some degree of renal failure is triggered, a phenomenon known as a drug-associated adverse event [27], so it can also induce nephrotoxicity during prolonged periods of administration.

To better understand liver and kidney function, different ratios were determined for each treatment group (Figure 2C), as described below [28,29].

#### 2.4.3. Determination of the Liver Injury Ratios

The AST/ALT, ALT/ALP, ALP/TB, and GGT/ALP ratios were calculated to assess liver injury [30,31,32,33]. The AST/ALT ratio (De Ritis ratio) was 1 ± 0.03 and 1.2 ± 0.07 for the KP and disease-free groups, respectively, indicative of stable liver function. The MTZ and Vh groups presented a value < 1, suggesting hepatocellular damage (due to ALT elevation). On the other hand, the LL group showed a value > 1, possibly due to fibrosis, intrahepatic cholestasis, or toxicity (*p* ≤ 0.05, ANOVA followed by Dunnett’s test) (Figure 2C) [30]. These histopathological findings and the AST/platelet (PLT) ratio index (APRI) were used to determine the degree of liver fibrosis in the groups (Figure 3B) [31]. The histopathological evaluation did not show the typical alterations associated with fibrosis. Moreover, the APRI was <0.5 in all groups, corroborating the lack of significant fibrosis based on histopathological evaluation [31]. The ALT/ALP ratio was lower in the KP (0.17 ± 0.01) and LL (0.25 ± 0.05) groups compared with the other treatment groups (*p* ≤ 0.05, ANOVA followed by the Tukey test) (Figure 2C). A low ALT/ALP ratio is commonly associated with drug-induced intrahepatic cholestasis, mainly when there is a preferential increase in ALP over GGT (Appendix A), as observed in the present study [30]. The ALP/TB ratio was >2 in all treatment groups compared with the disease-free group (2.23 ± 0.54) (Figure 2C). The ALP/TB and AST/ALT ratios are used to predict the risk of fulminant liver failure associated with adjacent pathology of various etiologies (infectious and non-infectious) [32]. The groups that showed a higher ALP/TB ratio compared with the disease-free group were LL (10.85 ± 1.28), KP (10.75 ± 3), and Vh (8.9 ± 2.9) (*p* ≤ 0.05, ANOVA followed by Dunnett’s test) (Figure 2C). Of these groups, only the LL group also presented an elevated AST/ALT ratio, which reinforces the diagnosis proposed in this work [32]. Finally, the GGT/ALP ratio was <5 in all groups, and the absence of obvious jaundice ruled out alcoholic liver disease (Figure 2C) [30,33].

#### 2.4.4. Determination of the ALA Prognosis Ratios

The AL/GGT, GGT/ALT, and AL/ALP ratios were used to predict the prognosis of ALA [34,35,36]. The AL/GGT ratio was <0.5 in all treatment groups, and they did not present significant differences among them (*p* ≥ 0.05, ANOVA followed by the Tukey test) (Figure 2C). Compared with the disease-free group, the KP (0.25 ± 0.18), LL (0.23 ± 0.038), and Vh (0.19 ± 0.05) groups showed a lower AL/GGT ratio. These differences were due to decreased albumin levels in the KP (1.7 ± 0.2 g/dL) and LL (1.7 ± 0.1 g/dL) groups (Appendix A). Of note, several investigations have demonstrated the predictive value of the AL/GGT ratio in various diseases [34]. However, the AL/GGT ratio should be interpreted with caution and solely based on liver function parameters and clinical correlations. Some studies with pyogenic liver abscesses have demonstrated that a low AL/GGT ratio is associated with a worse prognosis due to decreased AL and increased immunoglobulins [34]. This view could contradict the results from the present study. However, the low AL/GGT ratios in the KP and LL groups seem more related to liver or kidney dysfunction due to the toxicity associated with the drugs themselves, consistent with the other study results.

However, the AL/GGT ratio should be taken with caution and interpreted solely based on liver function parameters and clinical correlation. Some studies with pyogenic liver abscesses demonstrated that low AL/GGT values are associated with a worse prognosis due to decreased AL and increased immunoglobulins [34]. This statement could contradict the results observed in this study. But the low AL/GGT values in the KP and LL groups seem more related to liver or kidney dysfunctions due to the toxicity associated with the drugs themselves, according to the previous results of this study. The GGT/ALT ratio was similar for all treatment groups (Appendix A), except for the KP group (0.15 ± 0.081), which showed a significant difference compared with the Vh group (0.04 ± 0.01) (*p* ≤ 0.05, ANOVA followed by Dunnett’s test) (Figure 2C). Although it is not common to use the GGT/ALT ratio to evaluate the prognosis of an ALA [35], it is possible to predict that KP is a less effective treatment than LL or MTZ, although there were no significant differences with the disease-free group (*p* ≥ 0.05, ANOVA followed by Dunnett’s test).

The AL/ALP ratio was <0.1 in all treatment groups, lower than the value for the disease-free group (0.13 ± 0.023) (*p* ≤ 0.05, ANOVA followed by Dunnett’s test) (Figure 2C). The MTZ group showed greater similarity with the disease-free group, unlike the KP and LL groups (*p* ≤ 0.05, ANOVA followed by the Tukey test). The AL/ALP ratio has been used in oncology to assess disease progression [36], but it is not commonly used in the context of ALA progression. Nevertheless, it is possible to predict that MTZ may be more effective in combating an ALA than KP and LL.

#### 2.4.5. Determination of the Drugs Toxicity Ratios

The TB/AL ratio was used to determine the toxicity of the treatments [37]. It was significantly lower in the LL (0.22 ± 0.005) and KP (0.22 ± 0.04) groups (*p* ≤ 0.05, ANOVA followed by the Tukey test) compared with the disease-free group (0.40 ± 0.04), MTZ (0.38 ± 0.16), and Vh (0.48 ± 0.13) groups (Figure 2C). An elevated TB/AL ratio is associated with potential free (unconjugated or indirect) bilirubin neurotoxicity [38], indicating that the Vh group was at greater risk of this condition compared with the other treatment.

#### 2.4.6. Determination of the Kidney Damage Ratios

The DB/TB, BUN/CR, and urea/CR ratios were used to evaluate kidney damage or dysfunction [39,40,41,42]. The DB/TB ratio was <1 in all treatment groups, with no significant differences among them (*p* ≥ 0.05, ANOVA followed by the Tukey test) (Figure 2C). A high DB/TB ratio and high CR levels are associated with renal failure in some pathologies, such as malaria caused by *Plasmodium falciparum* [39]. Although the DB/TB ratio does not apply specifically to an ALA, the KP and LL groups were not at risk of developing hepatic or renal failure due to their similarities to the disease-free control. Thus, the BUN/CR and urea/CR ratios were used to assess possible causes of sudden (acute) kidney problems [40,41].

The BUN/CR and urea/CR ratios were <10 and <40, respectively, for the Vh group (7.9 ± 1.3 and 18.2 ± 1.35, respectively), which suggests intrinsic renal cause associated with the ALA (*p* ≤ 0.05, ANOVA followed by Dunnett’s test) (Figure 2C) [40,41]. The BUN/CR and urea/CR ratios were 19.6 ± 3.8 and 42 ± 8.15, respectively, for the MTZ group (*p* ≤ 0.05, ANOVA followed by Dunnett’s test), values that are associated with stable renal function or post-renal damage (Figure 2C) [41]. Therefore, these results do not adequately correlate with the preliminary findings regarding acute renal failure (Figure 3B). Note that the BUN/CR and urea/CR ratios for this group are within the upper limit of the considered normal range [42], and the treatment duration was relatively short. Finally, three groups presented a BUN/CR ratio > 25: KP (43.7 ± 4.3), LL (31.6 ± 9.8), and disease-free (34 ± 7.4) groups (*p* ≤ 0.05, ANOVA followed by Dunnett’s test) (Figure 2C). A value > 25 is associated with prerenal causes such as dehydration, congestive heart failure, and high protein consumption, among others [40]. Because the BUN/CR ratio was similar for the KP, LL, and disease-free groups (*p* ≥ 0.05, ANOVA followed by the Tukey test), it is possible to predict that the main cause is dehydration and to rule out any other underlying pathology [40]. Consistently, the urea/CR ratio was within the normal range of 40–110 for the KP (93.5 ± 9), LL (73.1 ± 26), and disease-free (112 ± 23) groups (*p* ≤ 0.05, ANOVA followed by Dunnett’s test) (Figure 2C), data that support dehydration [41].

#### 2.4.7. Evaluation of the Hematic Biometry

To conclude the paraclinical analyses, a complete blood count was performed to evaluate the evolution and resolution of the ALA (Figure 2D and Appendix A). The Vh group presented moderate leukocytosis (19,600 ± 4500/mm^3^) of the leukemoid type, without eosinophilia, and the presence of anemia (3.5 ± 0.4 × 10^6^/mm^3^) (*p* ≤ 0.05, ANOVA followed by the Tukey test) (Figure 2D and Table 3). The KP and LL groups did not present significant differences in their hematological parameters compared with the disease-free group (*p* ≥ 0.05, ANOVA followed by Dunnett’s test) (Figure 2D). The KP group presented values indicative of mild leukocytosis (14,800 ± 700/mm^3^) based on the reference values, but the LL and MTZ groups did not (*p* ≥ 0.05, ANOVA followed by the Tukey test) (Figure 2D). The results for the KP and LL may be related to remission of the ALA [43].

#### 2.4.8. Determination of the Hematic Ratios

The neutrophil/lymphocyte ratio is a reliable predictor of systemic inflammation and poor prognosis associated with complications from a pyogenic liver abscess [44]. The Vh group presented the highest neutrophil/lymphocyte ratio (2.2 ± 1.4), followed by the KP (1.67 ± 0.8), LL (1.66 ± 1.26), disease-free (1.30 ± 0.15), and MTZ (1.15 ± 0.7) groups. The similarities between the treatment groups rule out possible serious inflammatory processes and sepsis (*p* ≥ 0.05, ANOVA followed by the Tukey test). The MTZ group also presented a clinical picture of moderate anemia, similarly to what was observed for the Vh group, possibly due to side effects related to the treatment itself (*p* ≤ 0.05, ANOVA followed by Dunnett’s test) (Figure 2D) [42].

Other hematological parameters that tend to change during ALA progression but that did not show significant changes in the present study are the hemoglobin (Hb) level and the PLT count (generally due to mild to severe thrombocytopenia, without obvious bleeding) (*p* ≥ 0.05, ANOVA followed by the Tukey test) (Appendix A) [43,44]. On the other hand, there are some parameters that were not determined in this study and that can also be altered during ALA progression. They are related to inflammatory processes and include the erythrocyte sedimentation rate, the C-reactive protein level, the prothrombin time, and the fibrinogen level [43,44]. It would be interesting to analyze these parameters in subsequent studies.

### 2.5. Treatment with LL or KP Effectively Inhibited the Development of ALA in M. auratus

The magnetic resonance images (MRI) in Appendix A show the localization, morphology, and composition of the ALA in each group. In general, the ALA in the left hepatic lobe showed marked differences in morphology (ranging from ovoid with defined edges to irregular), extension (localized or infiltrative to the abdominal cavity), and volume (500–2000 mm^3^) (Table 2). However, most groups showed ovoid liver lesions with regular and well-defined borders. The internal appearance ranged from hypointense (T_1_) to heterogeneous (T_2_) in various areas, which aligns with the classifications proposed by several authors. Elizondo et al. [45] reported that an untreated ALA is associated with an incomplete annulus (corresponding to an incomplete wall) and diffuse or wedge-shaped perilesional edema, whereas a treated ALA is associated with complete ring formation and edema returning to form concentric rings around the abscess (clinical type: subacute mild ALA). These characteristics match with the ultrasound sonographic classification proposed by N’Gbesso [46] for the uncomplicated suppurative collection of an ALA (type II). To highlight the fluid component in the liver collections as hyperintense signals, the Short Tau Inversion Recovery (STIR) sequence was used after removing the adipose tissue signal (Appendix A). This sequence is relevant because the liquid content can be associated with inflammatory or liquefaction processes related to this pathology [18]. The order of hyperintense signals was the Vh group, followed by the MTZ, KP, and LL groups, a pattern that matches the extent of ALA progression (Appendix A). Furthermore, the MRI excluded common complications associated with the progression and dissemination of ALA to adjacent areas, such as pleural effusion, peritonitis, sepsis, subphrenic abscess, atelectasis, and pulmonary empyema [18].

The imaging results correlated with the histopathological evaluation of the livers and the ALA recovered from them. At the time of the necropsy, the diaphragmatic side of the liver showed simple, single grayish lesions due to invasion of amoebic trophozoites. These lesions contained thick, odorless, and sterile material due to acellular proteinaceous debris from necrotic liver tissue, as described in a previous ALA study [18]. However, only the Vh group presented discrete hepatomegaly compared with the disease-free group (*p* ≤ 0.05, ANOVA followed by Dunnett’s test) (Table 2). The weight and dimensions of the ALA collected from the treatment groups were significantly lower compared with the Vh group (*p* ≤ 0.05, ANOVA followed by Dunnett’s test) (Table 2 and Appendix A). The ALA volume was 19.42% smaller in the KP group compared with the Vh group and 28.16% smaller in the LL group compared with the Vh group; these findings are similar to the MTZ group (Table 2).

Based on histopathological evaluation, the Vh group presented alterations in the liver tissue architecture due to granulomas (Appendix A). These structures were of a variable size and contained necrotic-amorphous material, possibly due to contact cytolysis, degradative enzymes such as proteases, phagocytosis, and parasite resistance to host defense mechanisms [47]. Furthermore, capsule-shaped fibrosis, with compressed parenchyma, was delimited by amoebic trophozoites near the external wall of the ALA, similarly to a previous report [47]. Other characteristics included nonspecific periportal inflammation and the absence of amoebae within these areas (Appendix A). Similarly, the inflammatory infiltrate was a mixture of types of neutrophilic leukocytosis characterized by the presence of lymphocytes, plasma cells, and segmented neutrophils, which correlated with the blood count results (Table 3). On the other hand, the treatment groups exhibited similar histological characteristics, such as normal liver architecture but with vascular congestion, periportal inflammatory infiltrate, and the absence of amoebae within these areas (Appendix A). Finally, the LL and MTZ groups also showed an increase in hemorrhagic-type zones in many sinusoidal capillaries, characterized by an abundant content of red blood cells (Appendix A). During the amoebic invasive process, trophozoites can induce an inflammatory response in the liver (as an underlying effect of the disease) [48]. This process was observed in all groups (Appendix A). Moreover, the data suggest that KP and LL have an antiamoebic effect similar to that of MTZ, possibly promoting regeneration around the liver injury, although the injury was not fully resolved. However, additional in-depth studies are needed to corroborate this claim.

### 2.6. Toxicological Evaluation of Hamsters Treated with KP or LL

Morphometric analysis of the organs removed during necropsy was performed to observe changes in weight and dimensions associated with the treatments after infection (Table 4). The organs did not show significant morphological changes during anatomical inspection compared with those of the disease-free group (*p* ≥ 0.05, ANOVA followed by Dunnett’s test). The kidneys had the typical bean shape (with a convex area and other concave areas), with a capsule of whitish fibrous tissue interrupted at the level of the renal hilum, where the ureter and blood vessels are located (Figure 3A, kidneys). The heart was shaped such as an inverted cone, with the apex directed to the left. There were four cavities separated by a septum (atria and ventricles), and the vena cava (superior, inferior, and pulmonary) and other arteries (pulmonary and aortic) appeared at the base (Figure 3A, heart). The spleen was elongated, flat, thin, and curved at the edges, with a splenic hilum, numerous blood vessels, and a connective tissue capsule that gave off trabeculae inside (Figure 3A, spleen). The lungs were cone-shaped and divided into lobes. The lungs were located on the right and left within the thoracic cavity and showed some differences. The right lung (short and wide) had three lobes (upper, middle, and lower) divided by a horizontal and an oblique fissure. The left lung had a single lobe (Figure 3A, lungs).

Based on the histopathological evaluation, the organs maintained a normal appearance with the integrity of their tissue structure in the disease-free group and all treatment groups (Figure 3B). In general, the livers had a stroma of fibrous connective tissue and a parenchyma with numerous hepatocytes grouped in cell sheets. The liver lobes were not visible, and the central veins had a large diameter (Figure 3B, livers at 10× magnification). Additionally, there were portal spaces between nearby lobes, formed by portal triads comprising branches of the portal vein, the hepatic artery, and the bile duct (Figure 3B, liver at 40× magnification). These characteristics are consistent with those reported in the literature [49,50,51]. The liver tissue did not show any noticeable damage or change, except for those previously noted for this organ. However, there were hepatocytes of variable size and some eosinophilic (glycogenic) cells arranged in cords of two or more cells separated by sinusoids. These features are consistent with areas of possible regeneration (Figure 3C, letter H). Furthermore, the samples presented sinusoids lined with endothelial cells (Figure 3C, letter S), but with numerous Kupffer cells (Figure 3C, letter K), lymphocytes (Figure 3C, letter L), and vascular congestion associated with inflammatory and focal intrahepatic infiltrates. There was also subacute cholestasis (Figure 3C, gray arrow) due to the accumulation of discrete bile plugs in the bile canaliculi (Figure 3C, black arrow).

The kidneys had a capsule, a cortex, a medulla with numerous medullary pyramids, and a calyx. Many nephrons were observed, with a renal corpuscle and numerous tubules (Figure 3B, kidneys at 10× magnification). In turn, the glomeruli presented a proximal convoluted tubule, Bowman’s capsule, a mesangial matrix, a urinary space, capillary lumens, arterioles, and a distal convoluted tubule (Figure 3B, kidneys at 40× magnification). The kidneys did not show visible damage or relevant changes in tissue architecture, with some exceptions for the Vh and MTZ groups. Taken together, KP and LL administered for short times do not induce important damage to the liver of kidneys.

### 2.7. In Silico Analyses to Determine the Toxicity of KP and LL

Bioinformatics analysis of KP and LL was conducted to determine (i) whether they have pharmacological properties similar to those of antiprotozoal drugs commonly used against amoebiasis (such as MTZ); (ii) the possible adverse effects during treatment in an animal model; and (iii) the potential molecular targets of both compounds. These analyses revealed that KP and LL meet the criteria proposed by diverse drug-likeness models based on the physicochemical properties of both compounds (Figure 4 and Table 5), as follows: Lipinski (*log P*: ≤5, MW: ≤500 Da, HBA: ≤10, and HBD: ≤5) [52]; Ghose (*log P*: −0.4 to +5.6, MR: 40 to 130, MW: 180 to 480 Da, and A: 20 to 70) [53]; Veber (RB: ≤10, and PSA: ≤140 Å2) [54]; Egan (*log P*: ≤5.88, and PSA: ≤131.6) [55]; or Muegge (MW: 200 to 600 Da, *log P*: −2 to +5, PSA: ≤150, R: ≤7, C: >4, HA: >1, RB: ≤15, HBA: ≤10, and HBD: ≤5) [56].

The main physicochemical properties of KP are low flexibility and high unsaturation, medium polarity (*log P*: 1.61) with water solubility of 2.60 × 10^−4^ mol/L, and high passive absorption through the gastrointestinal tract (PSA: 87.13 A^2^ in CACO-2 cells) (Figure 4A). LL shows low flexibility, intermediate lipophilicity (*log P*: 2.73) with a water solubility of 1.33 × 10^−4^ mol/L, and high gastrointestinal absorption (PSA: 51.28 A^2^ in CACO-2 cells) and permeability passive across the blood-brain barrier (BBB score: 3.86) (Figure 4B). In both cases, the compounds have a high probability of being bioavailable at biological pH in rats (>10%, *F* = 0.55 based on the PSA values), and neither molecule appears to be permeable to the skin or eliminated from the central nervous system by *P*-glycoprotein (*P*-gp) (Figure 4D). Of note, *P*-gp is known as multidrug resistance protein-1 (MDR1, also known as ABCB1) and is abundant in the intestinal epithelium, liver, kidneys, BBB, blood-testicular barrier, and some tumors [57]. Therefore, KP and LL may be active against different types of tumors, but more in-depth studies are necessary to explore this possibility.

The drug-likeness model score (DLMS) of Molsoft© predicted that KP (DLMS: 0.5, high) is more similar to a drug compared with LL (DLMS: −0.55, low) (Figure 4A,B) and has higher bioactive potential with respect to MTZ (DLMS: 0.24, medium) (Appendix A). LL may have a low score due to the toxicity prediction associated with its molecular structure (ToxiM: 0.958) and structural alarm of Brenk (>2 esters) (Table 5). However, the experimental results showed that LL has higher selectivity in its biological effect compared to KP by different methods (Table 5 and Figure 4). Therefore, the DLMS results should be interpreted with caution and confirmed by results from an animal model.

The predicted biological activity in humans indicates that KP may act as a kinase inhibitor and nuclear receptor ligand, while LL may act as a ligand of G protein–coupled receptors (GPCRs) or nuclear receptor ligand and modulate ion channels (Figure 4). Among the most likely pharmacophore targets for KP and LL are NADPH oxidase IV (NOX4) and the kappa-opioid receptor (OPRK1) (Appendix A), respectively. NOX4 is mainly expressed in the kidney, and its function is to produce hydrogen peroxide (H_2_O_2_) [58]. OPRK1 is a receptor for opioid-like compounds in the brain, and its function is to regulate the effects of these compounds (such as alteration of nociception, consciousness, motor control, and mood) [59]. These findings are important because they allow for predicting anti-inflammatory properties for KP and sedative effects for LL, as well as the possible harmful effects of both compounds on human health and their probable antiparasitic mechanism against *E. histolytica*.

## 3. Discussion

This study provides evidence of the therapeutic potential of KP and LL as antiparasitic agents against *E. histolytica*. The focus was on predicting their mechanisms of action, cytotoxic potential in cell lines, and safety as treatments in an experimental animal model of ALA. The cell line results showed that *E. histolytica* is more sensitive to the antiparasitic effects of LL than KP. This may be attributed to the intrinsic chemical characteristics of LL as well as differences in how the two compounds interact with the parasite [9,10,11,12,13]. The cytotoxicity results revealed that both KP and LL induced apoptosis and necrosis in *E. histolytica* trophozoites [11,12], which was confirmed by phosphatidylserine externalization and alterations in cell membrane permeability. Although both compounds exhibited these effects, LL induced more pronounced necrosis, which could be related to its ability to trigger nonspecific inflammatory processes [13]. Cell cycle arrest and the alterations in the actin cytoskeleton observed in trophozoites treated with KP or LL represent another key mechanism of their action. The loss of critical structures such as filopodia, lamellipodia, and stress fibers in parasitic cells compromises their motility and invasive capacity, suggesting that these compounds might interfere with *E. histolytica* motility and its ability to invade and colonize extraintestinal body areas, potentially triggering the development of ALA and disease progression [2]. These findings were supported by the ultrastructural changes observed with TEM, reinforcing the potential of KP and LL as antiamoebic agents. However, it is worth noting that KP and LL showed a lower selectivity index (SI) compared with MTZ during evaluation of normal mammalian and human cell lines. This may result from differences in their mechanisms of action [9].

The in silico predictions suggested that KP and LL could be toxics. From a safety perspective, the results obtained from hamsters inoculated intrahepatically with *E. histolytica* trophozoites and then treated with KP or LL were promising. Neither compound induced significant toxicological effects or behavioral changes in the animals [60]. The treated hamsters did not exhibit signs of intoxication or severe clinical symptoms such as anasarca, anorexia, or abdominal pain, which are common in ALA progression models [18]. Although body temperature was not monitored, the absence of significant alterations in body weight and the 100% survival of the hamsters support the safety of these compounds. These findings are crucial as they suggest that KP and LL could be considered safe for future studies in preclinical models and eventually in humans.

The effects of KP and LL during ALA development were also evaluated based on biochemical and hematological parameters. The liver function parameters GGT, AL, ALP, AST, and ALT deviated from the reference data for hamsters in all groups, with some exceptions in the disease-free group [19,20]. The Vh, KP, MTZ, and LL groups presented significant hypoalbuminemia, hyperphosphatasemia, hyperbilirubinemia, and hypoazotemia. However, it is essential to emphasize that the most pronounced liver damage occurred in the control groups (Vh and MTZ). This suggests that some of the observed liver function alterations are not solely due to the action of KP and LL but rather to ALA progression and the effects of the standard treatment MTZ [2,18]. The MTZ is known to induce hepatotoxicity, which may explain the elevated transaminase levels and hyperphosphatasemia [5,6]. The correlation between histopathological and paraclinical findings supports the hypothesis that liver damage in the KP and LL groups was primarily related to ALA progression, while alterations in the MTZ group were likely a direct result of treatment toxicity [18].

The kidney profile showed an inverse trend compared with the liver profile, with decreased urea levels in the treated groups and a significant increase in the CR level in the MTZ group. This suggests that while KP and LL did not cause significant kidney damage, MTZ might induce early kidney dysfunction, consistent with its known renal elimination profile and potential nephrotoxicity [5]. Previous studies have shown that MTZ can accumulate in the body and alter glomerular filtration or tubular secretion [6], which might explain the nephrotoxic events observed in the present study.

Evaluation of relationships between hepatic and renal biochemical parameters provided additional insights into possible toxicity mechanisms. The De Ritis ratio [30] was <1 in the MTZ and Vh groups, indicating hepatocellular damage; the value >1 in the LL group could be related to toxicity or hepatic fibrosis. Additionally, the ALP/TB and GGT/ALP ratios confirmed possible intrahepatic cholestasis in the KP and LL groups, although at levels that do not suggest an immediate risk of fulminant liver failure. Additional studies are needed to determine whether prolonged exposure to these compounds could induce chronic hepatic complications, such as cholestasis [50,51].

Histopathological evaluation of liver tissue showed partial regeneration in the damaged areas in the KP and LL groups. This suggests that both compounds have an antiamoebic effect comparable with MTZ, although the observed regeneration was not sufficient to completely resolve the lesions. The presence of granulomas, necrosis, and fibrosis in the periportal areas of the liver in the treated groups confirms that while KP and LL limit damage induced by *E. histolytica*, they may not be entirely effective in advanced stages of the disease [2,18].

Finally, the prediction of biological activity in humans shows that KP could act as a kinase inhibitor and nuclear receptor ligand, while LL could be a ligand of GPCRs or nuclear receptors and modulate ion channels. These findings are relevant because they allow researchers to predict anti-inflammatory properties for KP and sedative effects for LL, as well as the possible harmful effects of both compounds on human health and their probable antiparasitic mechanism against *E. histolytica*. Several studies have shown that *E. histolytica* has homologous proteins similar to those mentioned above, and these proteins have relevant functions in the biology of this parasite, such as the ones presented below: (i) Kinases are abundant in this amoeba and can control cell functions such as cell signaling, metabolism, division, and survival [61,62]. (ii) Nuclear receptors are transcription factors that regulate the responses to small lipophilic compounds, and they play a role in every aspect of development, physiology, and disease in humans [63]. (iii) Signaling pathways associated with G-proteins are factors that contribute to the pathogenesis of this protozoan because they modulate amoebic motility through the regulation of the dynamic actin cytoskeleton, among other cellular processes (such as migration, vesicular trafficking in phagocytosis or the secretion of virulence factors, invasion, evasion of host immune response, and binding–killing of host cells) [64,65]. (iv) Ion channels are membrane proteins that regulate the passage of ions such as Ca^2+^, which in *E. histolytica* has cytopathic activity and a role in growth and differentiation [66]. Complementary studies are necessary to evaluate the role of the aforementioned proteins in the amoebicidal mechanism of action of KP and LL. For example, comparative studies could assess orthologous proteins from other amoebae (pathogenic and non-pathogenic) and their homologs in *Homo sapiens*.

Finally, the findings from the present study suggest that KP and LL have significant potential as antiamoebic agents. Future studies will be essential to delve deeper into specific molecular mechanisms and any potential adverse effects they may occur in more complex systems.

## 4. Materials and Methods

### 4.1. Chemical Compounds

The compounds evaluated in this study were KP (a tetrahydroxyflavone), purified from the aerial parts of *Cuphea pinetorum* Benth. (Common name: “cigar plant”; family: *Lythraceae*; The Plant List: KEW-2748311); and LL (a diterpene), purified from the aerial parts from *Salvia polystachya* Cav. (common name: “*chia*”; family: *Lamiaceae*; The Plant List: KEW-183536). Both compounds presented ≥95% purity and were obtained according to the procedure proposed by Calzada et al. [10,12]. MTZ was used as an antiparasitic reference in the different assays and acquired from Liomont Infinitely© Pharm (Flagenase 400, Ciudad de México, Mexico). The vehicle control was 1× PBS. The additional equipment and reagents used for the experiments are indicated in the text.

### 4.2. Cell Cultures

Cell lines were acquired from ATCC (Manassas, VA, USA), maintained at 80% confluence, and harvested in the log phase of growth for all experiments. The following transformed/non-tumorigenic cell lines were used: Chinese hamster ovarian epithelium (CHO-K1, CCL-61™) and human lung/bronchial epithelium (BEAS-2B, CRL-9609™). *E. histolytica* trophozoites (HM 1: IMSS, 30459™) were axenically cultivated in microaerophilic conditions, and virulence was maintained through successive passages into the liver of hamsters [67]. All cell lines were cultured and maintained according to the manufacturer’s instructions. The cell density at the time of collection was determined by trypan blue stain (0.4%, Sigma-Aldrich; St. Louis and Burlington, MA, USA). Cell viability, the IC_50_, SI, and LD_50_ were calculated as follows [17,68]: % Viability = (Abs_sample_/Abs_control_) *×* 100; IC_50_ was obtained by regression analysis (percent survival vs. log concentration); SI = IC_50_ normal cell line/IC_50_ trophozoites cell line; LD_50_ (oral, mg/kg) = 10^X^ [(0.372 × log IC_50_ normal cell line) + 2.024].

### 4.3. Cytotoxicity Assays in Normal Cell Lines by Formazan Salts

BEAS-2B and CHO-K1 cells were used to evaluate the cytotoxic effect of KP and LL with the 3-[4,5-dimethylthiazol-2-yl]-2,5-diphenyl tetrazolium bromide (MTT) assay. Cells (2.0 × 10^4^ cells/per well) were placed in 96-well microplates (Corning; Reynosa, TS, Mexico) with 200 µL of supplemented medium for 24 h. Then, the cells were treated with KP, LL, MTZ (10–500 μg/mL), or 1× PBS (Vh) for 24 h under the same conditions. Four hours before ending the treatments, 20 μL of MTT (5 mg/mL in 1× PBS, Sigma-Aldrich; St. Louis and Burlington, MA, USA) was added. Subsequently, the supernatant was removed, and the formazan was dissolved in 200 µL of acidified isopropyl alcohol (0.2% HCl, J.T. Baker; Radnor, PA, USA). The absorbance at 590 nm was read with a microplate reader (model 680, Bio-Rad; Hercules, CA, USA) to determine cell viability [69].

### 4.4. Cell Death Determination by Flow Cytometry

Based on the previously reported IC_50_, trophozoites (1 × 10^5^/mL) were cultivated and treated in the presence of compounds for 48 h. The treatments were MTZ (0.04, 0.08, or 0.16 μg/mL), KP (7.9, 15.8, or 31.6 μg/mL), LL (7.8 or 15.6 μg/mL), and Vh (1× PBS). Then, the trophozoites were resuspended in 500 μL of 1X binding buffer and stained with 5 μL of AnV-FITC and propidium iodide (PI) (K109, Biovin; Kedah, Malaysia) for 10 min at 37 °C. The samples were processed by flow cytometry on a BD FACS-Calibur^TM^ (Becton, Dickinson, and Company; Franklin Lakes, NJ, USA), and 20 × 10^3^ events were read at λ_ex/em_ 485/538 nm (AnV-FITC) and λ_ex/em_ 538/620 nm (PI). The obtained data (size, granularity, and fluorescence) were analyzed with the Flowing Software (version 2.5.1; Turku University, Turku, Finland) and simultaneously corroborated by confocal microscopy (LSM-700, Zeiss; Oberkochen, Germany) with a 40× objective lens [9].

### 4.5. Cell Morphology by Confocal Microscopy

A total of 1 × 10^5^/mL *E. histolytica* trophozoites were placed in sterile Lab-Tek™ chamber slides (Thermo-Scientific^TM^, Waltham, MA, USA), with 400 µL of supplemented medium (ATCC; Manassas, VA, USA), and treated with the IC_50_ of samples/controls for 1 h at 37 °C. Then, the culture medium was removed, and the adhered cells were fixed with 4% paraformaldehyde for 1 h at 37 °C. The samples were washed three times with 1× PBS and incubated with 1:50 phalloidin–rhodamine (R415, Invitrogen; Carlsbad, CA, USA) for 1 h at 37 °C. Finally, the cells were washed, mounted with VectaShield/DAPI medium (Vector Laboratories; Newark, CA, USA), and observed with a confocal microscope (LSM-700, Zeiss; Oberkochen, Germany) [9].

### 4.6. Ultrastructural Morphology Analysis with TEM

Trophozoites were treated with the IC_50_ of KP or LL for 48 h at 37 °C. Subsequently, they were harvested and processed for TEM. The trophozoites were fixed with 2.5% glutaraldehyde (Sigma-Aldrich; St. Louis and Burlington, MA, USA) in sodium cacodylate buffer (0.1 M, pH 7.2) for 24 h at 25 °C. Post-fixation was performed with 1% osmium tetroxide (Sigma-Aldrich; St. Louis and Burlington, MA, USA) for 1 h at 25 °C. The samples were dehydrated with ethanol:propylene oxide (Sigma-Aldrich; St. Louis and Burlington, MA, USA) (50, 70, 90, and 100% ethanol) for 10 min at 4 °C and placed in polymerized epoxy resin (Poly/Bed 812, Polysciences; Warrington, PA, USA) at 60 °C for 24 h. Ultrathin sections (60 nm thick) were obtained with an ultramicrotome (Porter-Blum MT-1, Sorvall, MA, USA) and stained with 2% uranyl acetate (Polysciences; Warrington, PA, USA) for 20 min and 0.2% lead citrate (Polysciences; Warrington, PA, USA) for 5 min. Finally, the samples were observed with a TEM (JEM-1011, JEOL; Tokyo, Japan) [9,70].

### 4.7. Hamster Model for ALA

*M. auratus* was used as an animal model because the hamster is naturally susceptible to infection by *E. histolytica* and can develop amoebic colitis or extraintestinal complications such as an ALA [71]. In this context, the hamster can be used to search for new alternative therapeutic agents against amoebiasis [16]. The hamsters were acquired from Charles River Laboratories Inc. (Wilmington, MA, USA) in 2017 by Centro de Investigación y de Estudios Avanzados del Instituto Politécnico Nacional (CINVESTAV-IPN). The supplier’s health reports indicated that the hamsters were free of known viral, bacterial, and parasitic pathogens. Male hamsters (sexually mature and homozygous) weighing 95 ± 5 g and ≈10 weeks old were selected for the experiments. They were maintained in aseptic conditions at 25 ± 1 °C, 50 ± 3% humidity, a 12 h photoperiod, and ad libitum access to a standard diet (LabDiet; Richmond, IN, USA) and water (Figure 5A). The hamsters were housed in polycarbonate boxes (five per box), with an American Iron and Steel Institute (AISI) lid and wood shavings as bedding, in a controlled room of the Animal Production and Experimentation Unit (UPEAL) from CINVESTAV-IPN (Figure 5A). Additionally, the health state of the hamsters and their adaptation to the new laboratory conditions were evaluated for 15 days to decrease their stress and anxiety levels. This study was carried out according to the Official Mexican Regulations (NOM-062-ZOO-1999) and NC3Rs ARRIVE [60,72]. The protocol was approved by the Institutional Animal Care and Use Committee (Reg. No. 0183-16). Finally, the hamsters were not subjected to any previous experimental procedure or any additional treatment before beginning the experimental procedures (see Appendix A, NC3Rs ARRIVE Guidelines Checklist). Compounds that are prohibited and/or not recommended for use in animals by the American Veterinary Medical Association (Guidelines for the Euthanasia of Animals) were not used [73].

### 4.8. ALA Induction and the Antiparasitic Protocol

These experiments were performed with three independent biological replicates. For each replicate, twenty-five hamsters were randomly distributed (simple type) into five experimental groups (five animals per group). The sample size was determined based on the Organization for Economic Cooperation and Development (OECD) recommendations that establish the minimum number of individuals required for the tests so as not to sacrifice animals unnecessarily [74]. Laparotomy was performed in aseptic conditions by using depilation, iodine solution, as well as sterile field cloths. Subsequently, an intra-hepatic puncture was made to deliver 1.5 × 10^6^ virulent *E. histolytica* trophozoites in 100 μL of 1× PBS into the left hepatic lobe for ALA induction [18] (Figure 5A-3). The injection site was immediately covered by applying a gel foam pad to maintain hemostasis. The abdominal layers were sutured discontinuously by single vertical stitches using Vicryl synthetic absorbable suture with a triangular pointed half circle needle. Irregular edges or folds in the surgical wound were eliminated to avoid tissue devitalization or possible infections [67]. The development of a hepatic ALA was monitored for 4 days (Figure 5A-4).

The antiparasitic activity protocol started with the administration of treatment on day 4 post-infection, with i.p. delivery every 24 h for 4 days, to favor greater bioavailability of the compounds during the experimental model [71] (Figure 5B). The compounds were administered by the i.p. route because the peritoneal cavity presents rapid fluid absorption into the bloodstream, and initial biotransformation by the gastrointestinal tract is avoided [75]. The optimal doses of KP and LL were determined based on previous cytotoxic studies, bioinformatics analysis, and the MTZ doses used in humans. Table 6 provides more detail on the doses used in this study. The hamsters were evaluated 24 h post-treatment to detect any signs of discomfort/pain or toxicity. The humane endpoint criteria used in this study were >20% body weight loss overnight, cachexia, recumbence, lethargy, respiratory distress, inability to reach food and water, lack of grooming, and ulceration around the surgical site, among others [76]. Additionally, the weight of hamsters was determined at the beginning and end of the experiment with an electronic bascule (CS200, Ohaus; Parsippany, NJ, USA), and the ALA volume was calculated based on imaging (Figure 5B-1). Sodium pentobarbital for veterinary use (Pisabental of 6.3 g/100 mL, PiSA; Monterrey, NL, Mexico) was applied as an anesthetic (95 mg/kg body weight) [60,73] for xenotransplantation (a single dose, Figure 5A-3) and prior to each paraclinical evaluation (a single dose for each evaluation). A single lethal dose of the same anesthetic (285 mg/kg body weight) was also used for euthanasia. In all cases, the anesthetic was administered by i.p. injection, with the hamster placed in the supine position by grasping the nape and the skin fold between the lower/middle back. The administration of analgesics was not necessary. Biological samples—blood, organs, and abscesses—were obtained post-mortem (Figure 5B). Animal carcasses were placed in a yellow polyethylene bag to discard pathological waste, stored at 4 °C, and then transported to a biohazardous and infectious waste collection center for incineration [77].

### 4.9. MRI

MRI analysis was performed using a MAGNETOM Symphony with a Tim 1.5 Techo system (Siemens Medical Solutions; Forchheim, Germany) with a human knee diagnostic antenna without paramagnetic contrast. The hamsters were placed in supine and craneocaudal positions for imaging analyses. The sequences/parameters used were: coronal projections T_1_ (thickness of cut: 1.5 mm; TR/TE: 500/20 ms; field of view: 130 mm; matrix: 512), T_2_ (thickness of cut: 2 mm; TR/TE: 3890/117 ms; field of view: 180 mm; matrix: 512), and STIR (thickness of cut: 3 mm; TR/TE/TI: 5000 ms/29 ms/130 ms; field of view: 153 mm; matrix: 512). After imaging, the results were analyzed in the RadiAnt DICOM Viewer software (v3.4, Medixant; https://www.radiantviewer.com (accessed on 14 September 2024)) to perform the measurement and characterization of ALA lesions, as well as three-dimensional reconstructions [9].

### 4.10. Paraclinical Analysis

Peripheral blood samples were obtained from anesthetized hamsters via retro-obital puncture using heparinized capillary tubes (Vitrex; Herlev, Denmark) and collected in pediatric tubes with K_2_EDTA (BD Microtainer; Becton, Dickinson and Company, Franklin Lakes, NJ, USA) for hematic biometry or without anticoagulant for chemical sanguineous (BD Microtainer; Becton, Dickinson and Company, Franklin Lakes, NJ, USA). Serum was obtained by centrifugation at 3500 rpm and frozen at −70 °C until analysis. Hematic biometry was performed in a hematology autoanalyzer system (BC-2300, Mindray; Shenzhen, China), and biochemical parameters were made on an automatic medical system (Prestige 24i, Tokyo Boeki; Kanagawa, Japan) [9,70].

### 4.11. Histopathological Analysis

After blood samples were obtained, the hamsters were sacrificed with a lethal dose of sodium pentobarbital, followed by a cervical fracture. They were mounted on a dissecting board for the necropsy. A complete mid-laparotomy was performed to remove the lungs, spleen, heart, liver, and kidneys. An excisional biopsy was performed to recover the ALA lesions for morphometric analysis. The organs and ALA lesions were rinsed with 1× PBS, weighed, and adherent tissue was removed. Subsequently, samples were fixed in 4% paraformaldehyde (Sigma-Aldrich; St. Louis and Burlington, MA, USA) and stored at 4 °C for 48 h. Later, the inclusion protocol was performed to obtain histological sections (3 μm thick) with a rotatory microtome (RM2125-RTS, Leica; Wetzlar, Germany). Tissue slices were stained with hematoxylin-eosin (Merck; Rahway, NJ, USA), mounted on coverslips with Entellan^TM^-New (Merck; Rahway, NJ, USA), and observed with optical microscopy (Eclipse 80i, Nikon; Tokio, Japan) [9,70].

### 4.12. In Silico Analyses

The chemical structures, molecular properties, and Simplified Molecular-Input Line-Entry System (SMILES) notations of the compounds were obtained by ChemSketch^TM^ (v1 2.0, ACD; Toronto, ON, Canada). The SMILES notations for KP and LL were fed into the SwissDrugDesign© applications (Swiss Institute of Bioinformatics; Lausanne, Switzerland) (https://www.molecular-modelling.ch/swiss-drug-design.html, accessed on 20 April 2023), including (i) SwissADME (v2019, SIB©), to create predictive models for the physicochemical properties, pharmacokinetics, drug likeness, and medicinal chemistry friendliness of multiple molecules [78]; and (ii) SwissTargetPrediction (v2017, SIB©), to predict possible targets of bioactive small molecules in humans and other vertebrates [78]. Additionally, to complement and validate the data provided by SwissDrugDesign©, the following online bioinformatic programs were used: (i) Molsoft Drug-Likeness model (Molsoft; San Diego, CA, USA) (http://molsoft.com/mprop/, accessed on 20 April 2023); (ii) Molinspiration© (v2018.10, Molinspiration Cheminformatics©; Bratislava, Slovak Republic) (www.molinspiration.com (accessed on 14 September 2024)); (iii) ToxiM (IISER©; Bhopal, India) (http://metabiosys.iiserb.ac.in/ToxiM, accessed on 20 April 2023) [79]; (iv) T.E.S.T. (v4.1, U.S. EPA) (https://comptox.epa.gov/dashboard/predictions/index, accessed on 20 April 2023) [80]; (v) Toxtree (v3.1.0, Ideaconsult Ltd.; Sofia, Bulgaria) (http://toxtree.sourceforge.net/, accessed on 20 April 2023); (vi) SuperCYPsPred (v2020, Structural Bioinformatics; Berlin, Germany) (http://insilico-cyp.charite.de/SuperCYPsPred/index.php?site=home, accessed on 20 April 2023) [81]; and (vii) LAZAR (v1.4.2, IST-GmbH; Freiburg, Germany) (https://lazar.in-silico.ch/, accessed on 20 April 2023) [82].

### 4.13. Statistical Analysis

The results are presented as follows: for the in vitro studies, the mean ± standard deviation of three independent assays performed in triplicate (*n* = 9); for the in vivo studies, the mean ± standard deviation of two biological replicates (*n* = 5). When other descriptive statistics were used, such as the median and range, they are specified in the text or figure. The Anderson–Darling and Grubbs tests were used to assess normality and outliers, respectively. The results were analyzed with Student’s *t*-test or one-way ANOVA (for parametric data with a normal distribution), followed by pairwise comparisons with the Tukey–Kramer test or Dunnett’s test. Minitab (v18.1.0 for Windows, Minitab^®^ Statistical Software; State College, PA, USA) and GraphPad Prism (v8.0.2 for Windows, GraphPad Software; Boston, MA, USA; www.graphpad.com) were used for the statistical analysis. A significant *p*-value is indicated with * (≤0.05) or *** (<0.01).

## 5. Conclusions

The results of this study demonstrated the antiparasitic effects of KP and LL on *E. histolytica* trophozoites, evidenced by a reduction in cell viability, alterations in mobility, and morphological changes associated with cell death, including apoptosis, cytoskeletal changes, and cell cycle arrest. However, these effects were not highly selective, with *E. histolytica* being more sensitive to LL than to KP. Bioinformatics predictions indicated that KP might act as a kinase inhibitor and nuclear receptor ligand, while LL could function as a GPCR ligand and modulate ion channels. Additionally, KP and LL demonstrated antiparasitic effects in hamsters infected with *E. histolytica* trophozoites, as shown by a reduction in ALA lesions, supported by imaging and histopathological evaluations. However, KP was less effective than LL, and toxicological evaluation revealed that LL treatment was more toxic than KP treatment. These findings highlight the therapeutic potential of KP and LL against parasitic diseases related to *E. histolytica*, but additional in-depth research is necessary to understand their potential adverse effects on human health and to elucidate their antiparasitic mechanisms.

## Figures and Tables

**Figure 1 ijms-25-10633-f001:**
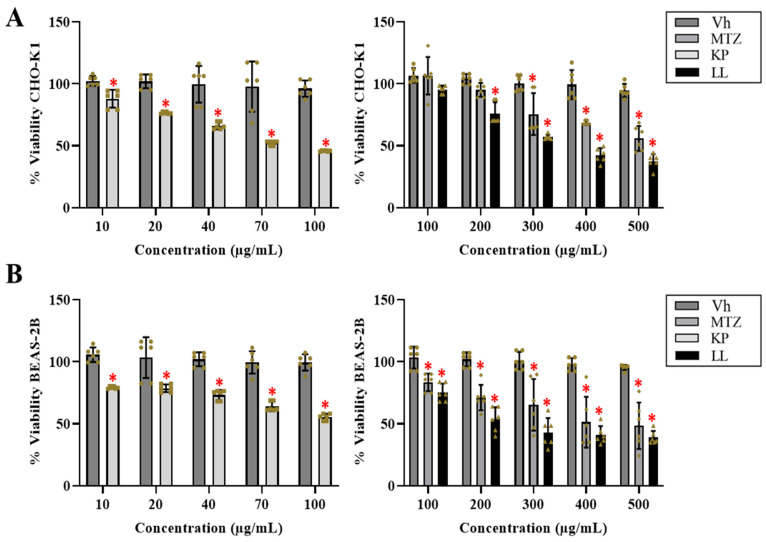
The cytotoxic effects of kaempferol (KP) and linearolactone (LL) on the CHOK-1 and BEAS2B cell lines. The half-maximal inhibitory concentrations (IC_50_) of KP and LL in the CHO-K1 (**A**) and BEAS-2B (**B**) cell lines were determined by dose-response viability curves after 24 h using the MTT assay. The results are presented as the mean ± standard deviation of three biological replicates (*n* = 3, in triplicate). (*) *p* ≤ 0.05 vs. vehicle (1× phosphate-buffered saline, negative control) by analysis of variance. Metronidazole (MTZ) was used as a positive control (antiparasitic reference).

**Figure 2 ijms-25-10633-f002:**
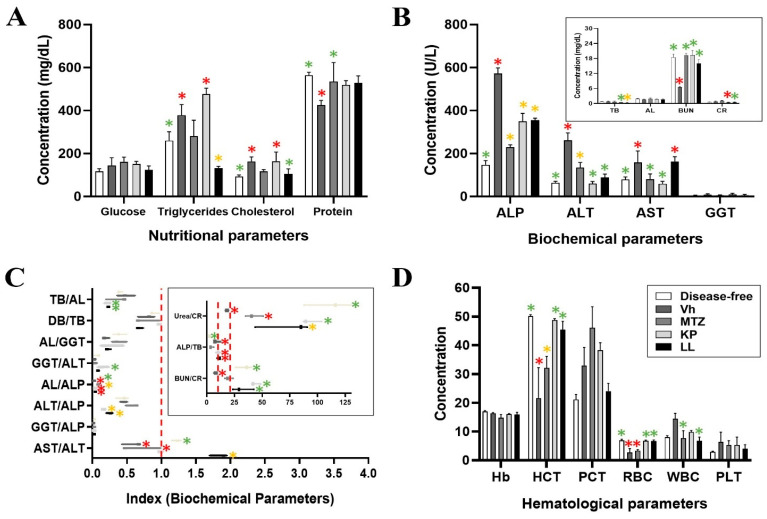
Paraclinical analyses of blood and serum from the five groups of hamsters. Nutritional parameters (**A**), biochemical parameters (**B**), ratios (**C**), and hematological parameters (**D**) were evaluated (for details, see Appendix A). The results are presented as the mean ± standard deviation (**A**,**B**,**D**) or the median and range (**C**) of two biological replicates (*n* = 5). * *p* ≤ 0.05 vs. the control group by analysis of variance (red for the disease-free group, green for the Vh group, and yellow for both groups). Metronidazole (MTZ) was used as a positive control, vehicle (Vh, 1× phosphate-buffered saline, no treatment) was used as a negative control, and disease-free (without ALA) was used as a normal control. ALP, alkaline phosphatase; ALT, alanine aminotransferase (GPT); AST, aspartate aminotransferase (GOT); GGT, gamma glutamyl-transpeptidase; TB, total bilirubin; AL, albumin; BUN, blood urea nitrogen; CR, creatinine; DB, direct bilirubin; Hb, hemoglobin (g/dL); HCT, hematocrit (%); PCT, plateletcrit (%); RBC, red blood cells (×10^6^/mm^3^); WBC, white blood cells (×100/mm^3^); PLT, platelets (×10^6^/μL).

**Figure 3 ijms-25-10633-f003:**
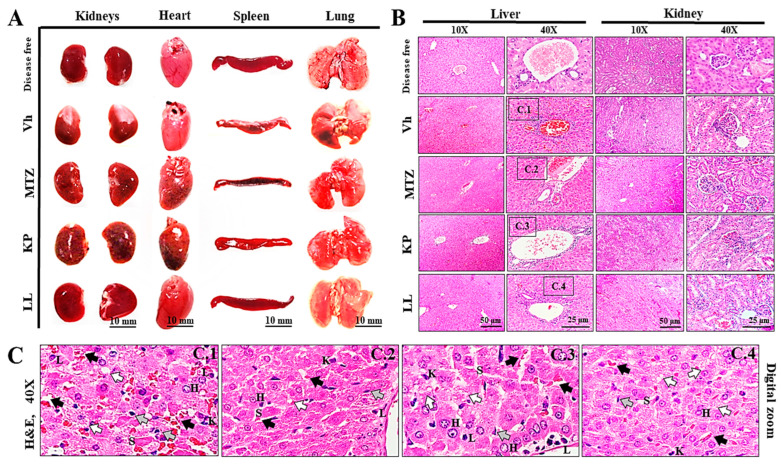
Histopathological analysis of the organs collected from the five groups of hamsters. The photographs show the anatomical morphology of the kidneys, heart, spleen, and lungs collected from the hamsters (**A**). Representative liver and kidney sections stained with hematoxylin and eosin are shown at 10× and 40× magnification (**B**). Digital zoom with 40× magnification of liver sections stained with hematoxylin and eosin clearly shows the following tissue characteristics: hepatocytes (H), sinusoids (S), Kupffer cells (K), lymphocytes (L), focal intrahepatic cholestasis (gray arrow), bile canaliculi (black arrow), and lipid microvesicles (white arrow) (**C**). The representative results from two biological replicates (*n* = 5) are shown. KP, kaempferol; LL, linearolactone. Metronidazole (MTZ) was used as a positive control, vehicle (Vh, 1× phosphate-buffered saline) was used as a negative control, and the disease-free status was used as a normal control.

**Figure 4 ijms-25-10633-f004:**
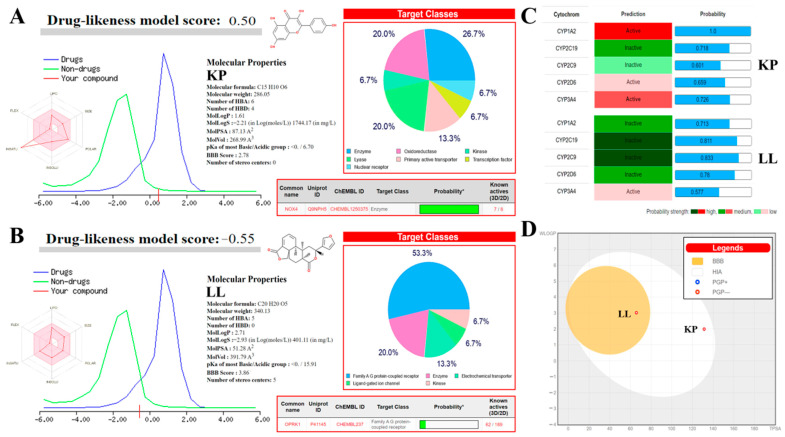
Predictions of the pharmacokinetics, toxicity, and molecular targets of KP and LL based on their physicochemical properties. The molecular properties and chemical structures of (**A**) KP and (**B**) LL were used to predict similarities with biologically active drugs through the Molsoft© drug similarity model and their potential target pharmacophores using SIB© SwissTarget prediction. (**C**) The toxicological activity of both active principles, similarly to antipara-sitic drugs in humans, was predicted through inhibitory effects on the enzyme complex associated with CYP using a predictive machine learning model based on specific fingerprints called MCF with SuperCYPsPred of Structural Bioinformatics©. (**D**) Other toxicity characteristics, such as BBB permeability, HIA, and PGP affinity, were predicted by the SIB© SwissADME boiled egg permeation method. The results were compared with metronidazole (MTZ), an amoebicidal compound used to treat diverse parasitic diseases (see Appendix A). Abbreviations used: KP, kaempferol; LL, linearolactone; MCF, most common features; BBB, blood-brain barrier; HIA, health impact assessment; PGP, *P*-glycoprotein; CYP, cytochrome-P450 system; HBA, hydrogen bond acceptors; HBD, hydrogen bond donors; *log P*, partition coefficient; *log S*, aqueous solubility coefficient; *pKa*, acid dissociation coefficient; PSA, polar surface area; LIPO, lipophilicity; POLAR, polarity; INSOLU, insolubility; INSATU, insaturation; FLEX, flexibility, NOX4, NADPH oxidase IV; OPRK, Kappa Opioid receptor.

**Figure 5 ijms-25-10633-f005:**
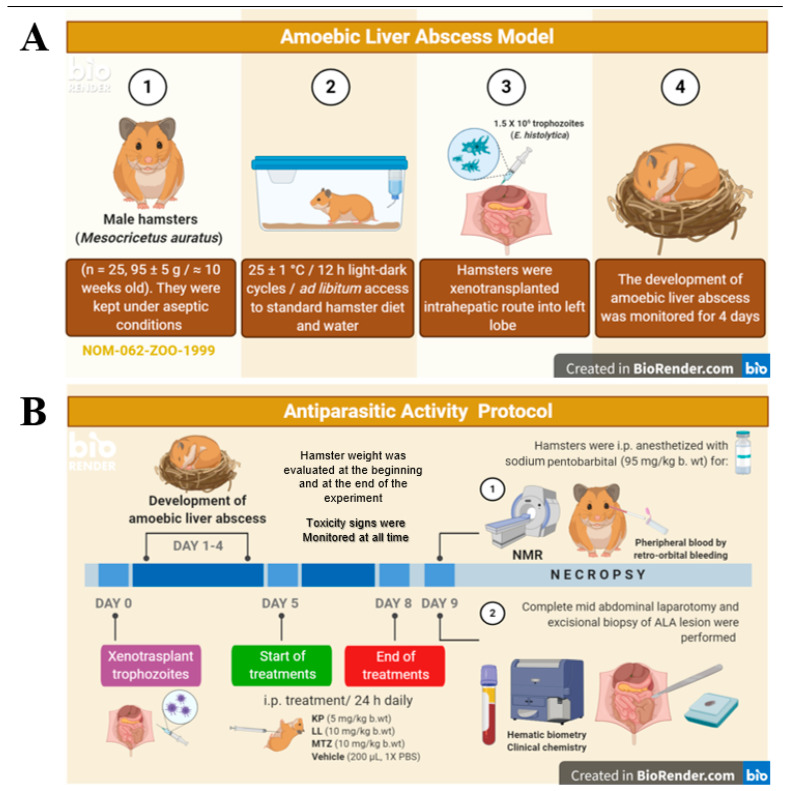
Timeline of the amoebic liver abscess model (**A**) and antiparasitic activity protocol (**B**) implemented for KP and LL. The antiprotozoal activity of kaempferol (KP) and linearolactone (LL) was determined in *Entamoeba histolytica* trophozoites xenotransplanted into the livers of male *Mesocricetus auratus* (Syrian hamsters), according to the protocol proposed by Tsutsumi et al. [18] and the Official Mexican Regulations (NOM-062-ZOO-1999) [72]. Figure created in BioRender by Talamas-Rohana, P. (accessed on 14 September 2024).

**Table 1 ijms-25-10633-t001:** Toxicological activities of kaempferol (KP) and linearolactone (LL) in the CHOK-1 and BEAS2B cell lines.

In Vitro Studies: IC_50_, SI and LD_50_				
Samples	Vh	MTZ	KP	LL
IC_50_ HM-1	Innocuous	0.16 ^++^ *	31.6 ^+^ *	<7.8 ^+^ *
IC_50_ CHO-K1	Innocuous	500 ^-^ *	70 ^+/-^ *	300 ^-^ *
IC_50_ BEAS-2B	Innocuous	400 ^-^ *	100 ^+/-^ *	200 ^-^ *
SI	ND	3461 ^++^	2.7 ^+/-^	>32 ^+^
LD_50_ (Theoretical)	ND	1025.64	551.75	824.20

The symbols correspond to the biological activity of the samples: ^++^, very high; ^+^, high; ^+/-^, medium; ^-^, low. The selectivity index (SI) indicates specific activity in normal cells vs. trophozoites (µg/mL). IC_50_, half-maximal inhibitory concentration (µg/mL). LD_50_, median lethal dose (oral, mg/kg). ND, not determined. The results are presented as the mean ± standard deviation of three biological replicates (*n* = 3, in triplicates). (*) *p* ≤ 0.05 vs. Vh (vehicle, 1× phosphate-buffered saline, negative control), based on analysis of variance. Metronidazole (MTZ) was used as a positive control (antiparasitic reference).

**Table 2 ijms-25-10633-t002:** The antiparasitic effect of kaempferol (KP) and linearolactone (LL) on the development of an amoebic liver abscess (ALA).

In Vivo Studies: Acute Toxicity and ALA Characteristics				
Treatments	Vh	MTZ	KP	LL
Hamster weight (g)	91.9 ± 2.3	95.4 ± 5.1	93.5 ± 6	93.4 ± 7.2
Toxicity signs	None	None	None	None
Survival rate (%)	100	100	100	100
ALA Weight (g)	1.23 ± 0.15	0.33 ± 0.32 *	0.86 ± 0.15 *	0.40 ± 0.20 *
ALA Volume (mm^3^)	2.257 ± 523.7	528 ± 183.5 *	738 ± 150 *	562.3 ± 23.3 *
MRI (T_1_/T_2_)	Intermediate intensity/ hyperintense	Intermediate intensity/ hypointense	Intermediate intensity/ hypointense	Intermediate intensity/ hypointense
Leukocyte infiltrate	++	−	+/−	+/−
Fibrosis	+	−−	−−	−−
Liquid	++	−	+/−	−
Edges	Irregular and poorly defined	Regular and well defined	Irregular and poorly defined	Regular and well defined
Morphology	Loculated ovoid	Round	Ovoid	Loculated ovoid

The following symbols correspond to the biological activity of the samples: ++, very high; +, high; +/−, medium; −, low; and −−, very low. T_1_, longitudinal relaxation times; T_2_, transverse relaxation times. The results are presented as the mean ± standard deviation of two biological replicates (*n* = 5). (*) *p* ≤ 0.05 vs. Vh (vehicle, 1× phosphate-buffered saline, negative control), based on analysis of variance. Metronidazole (MTZ) was used as a positive control (antiparasitic reference).

**Table 3 ijms-25-10633-t003:** Hematic biometry of hamsters treated with KP or LL by manual differential analysis.

Treatments (%) Hematological Parameters	Disease-Free	Vh	MTZ	KP	LL	Reference Range (Mean)
Lymphocytes	42 ± 4	**35 ± 13**	50 ± 13	**39 ± 12**	44 ± 21	40–85 (63)
Monocytes	0	0	0	0	0	1–6 (3)
Eosinophils	0	0	0	0	0	1–2
Basophils	0	0	0	0	0	0–5 (2)
Segmented neutrophils	54 ± 2	** 64 ± 14 **	51 ± 13	** 60 ± 11 **	** 56 ± 21 **	25–55 (40)
Band neutrophils	**4 ± 2**	**2 ± 1**	0	**2 ± 1**	0	5–13 (9)
Immature forms	0	0	0	0	0	0

**Bold** letters indicate differences with the reference range, and underlined letters indicate values outside of this limit. The reference range is presented as the minimum and maximum normal values for the analyte and the mean [27]. The results are presented as the mean ± standard deviation of two biological replicates (*n* = 5). *p* ≤ 0.05 vs. the vehicle (Vh, 1× phosphate-buffered saline, negative control) by analysis of variance. Metronidazole (MTZ) was used as a positive control (antiparasitic reference). Abbreviations used: KP, kaempferol; LL, linearolactone.

**Table 4 ijms-25-10633-t004:** Morphometric analysis of the organs collected from the five groups of hamsters.

Organs	Disease-Free	Vh	MTZ	KP	LL
Liver	6.7 ± 0.07/4.04 ± 0.05	**8.28 ± 1.1** */**4.7 ± 0.5** *	5.76 ± 1.4/4 ± 0.15	5.9 ± 0.8/3.88 ± 0.11	5.5 ± 0.9/3.94 ± 0.16
Heart	0.4 ± 0.0/0.8 ± 0.09	0.38 ± 0.05/1.06 ± 0.09	0.38 ± 0.1/1.08 ± 0.04	0.38 ± 0.1/1.06 ± 0.08	0.38 ± 0.08/1.1 ± 0.07
Kidneys	0.65 ± 0.07/1.2 ± 0.14	0.62 ± 0.03/1.2 ± 0.07	0.52 ± 0.0/1.3 ± 0.08	0.49 ± 0.04/1.2 ± 0.1	0.49 ± 0.07/1.1 ± 0.08
Lungs	0.67 ± 0.0/2.8 ± 0.2	0.7 ± 0.05/2.6 ± 0.2	0.68 ± 0.05/2.5 ± 0.2	0.74 ± 0.06/2.5 ± 0.05	0.74 ± 0.06/2.4 ± 0.1
Spleen	0.2 ± 0.0/3.4 ± 0.38	0.2 ± 0.0/3.30 ± 0.4	0.2 ± 0.05/3.3 ± 0.3	0.36 ± 0.05/3.7 ± 0.1	0.26 ± 0.05/3.4 ± 0.4

The table presented the weight (g)/largest diameter (cm) for each organ in the different treatment groups. **Bold** letters indicate significant differences with respect to the disease-free group. The results are presented as the mean ± standard deviation of two biological replicates (*n* = 5). (*) *p* ≤ 0.05 vs. the vehicle (Vh, 1× phosphate-buffered saline, negative control) by analysis of variance. KP, kaempferol; LL, linearolactone. Metronidazole (MTZ) was used as a positive control (antiparasitic reference).

**Table 5 ijms-25-10633-t005:** Bioinformatic analysis of pharmacological and toxicological properties of KP and LL.

Samples	MTZ	KP	LL
**SwissADME©: Physicochemical Properties**			
Density:	1.5 ± 0.1 g/cm^3^	1.7 ± 0.1 g/cm^3^	1.3 ± 0.1 g/cm^3^
Refraction index:	1.612	1.785	1.612
Polarizability:	16.2 ± 0.5 × 10^−24^ cm^3^	28.3 ± 0.5 × 10^−24^ cm^3^	35.0 ± 0.5 × 10^−24^ cm^3^
Surface tension:	60.5 ± 7.0 dyne/cm	98.9 ± 3.0 dyne/cm	54.6 ± 5.0 dyne/cm
Number heavy atoms:	12	21	25
Number of bonds:	12	23	29
Number of rings:	1	3	5
Number aromatic heavy atoms:	5	16	5
Fraction Csp3:	0.50	0.00	0.50
Number rotatable bonds:	3	1	1
Total charge:	0.0	0.0	0.0
Molar refractivity:	43.25 Å	76.01 Å	88.42 Å
**SwissADME©: Pharmacokinetics/Molinspiration©: Bioactivity score**			
GI absorption:	High	High	High
*P*-gp substrate:	No	No	No
*Log Kp*:	−7.36 cm/s	−6.70 cm/s	−6.37 cm/s
GPCR ligand:	−1.09	−0.10	**0.65**
Ion channel modulator:	−0.87	−0.21	**0.16**
Kinase inhibitor:	−0.59	**0.21**	−0.13
Nuclear receptor ligand:	−1.74	**0.32**	**0.66**
Protease inhibitor:	−1.68	−0.27	0.04
Enzyme inhibitor:	−0.32	**0.26**	**0.47**
**SwissADME©: Medicinal Chemistry**			
PAINS:	0 alert	**1 alert (catechol A)**	0 alert
Brenk:	**2 alerts (nitro group)**	**1 alert (catechol A)**	**1 alert (>2 esters)**
Leadlikeness:	**No (M.W. < 250)**	Yes	Yes
Synthetic accessibility:	2.30	3.14	5.56
**T.E.S.T.© and LAZAR©: Toxicological properties**			
LD_50_ *Fathead minnow* (96 h):	424.1 mg/L	1.28 mg/L	ND
LD_50_ *Daphnia magna* (48 h):	39.14 mg/L	3.62 mg/L	ND
IGC_50_ *T. pyriformis* (48 h):	270.22	10.54 mg/L	ND
LD_50_ Rat (Oral):	2444	2018 mg/kg	ND
Bioconcentration factor:	1.914	**8.032**	ND
Developmental toxicity:	Not	**Yes**	ND
AMES mutagenicity:	**Yes (*p* = 0.67)**	**Yes (*p* = 0.42)**	ND
Carcinogenicity (rodents):	ND	No (*p* = 0.43)	ND
Adverse effects (rat):	ND	1320 mg/kg/day	ND
Estrogen Receptor RBA:	5.089 × 10^−4^	0.004	ND
Estrogen Receptor Binding:	**Yes**	**Yes**	ND

The results shown in bold are out-of-range values. ND, not determined; ADME, absorption, distribution, metabolism, and excretion; GI, gastrointestinal; *P*-gp, *P*-glycoprotein; *Log Kp*, human skin permeability coefficients; GPCR, G protein-coupled receptor; PAINS, pan assay interference structures; Brenk, structural alert; LD_50_, median lethal dose; IGC_50_, half-maximal inhibition growth concentration; RBA, relative binding affinity; KP, kaempferol; LL, linearolactone. Metronidazole (MTZ) was used as a positive control (antiparasitic reference).

**Table 6 ijms-25-10633-t006:** Treatment scheme used in this study of the antiparasitic activity of KP and LL.

Groups	*n*	Treatment	Administration Route	Doses/Time
I (normal control, without ALA)	5	Disease-free	i.p.	N/A
II (negative control, non-treatment)	5	Vh (1× PBS)	i.p.	200 µL/day
III (positive control, antiamoebic drug)	5	MTZ	i.p.	10 mg/kg body weight/day
IV (sample)	5	KP	i.p.	5 mg/kg body weight/day
V (sample)	5	LL	i.p.	10 mg/kg body weight/day

Abbreviations used: KP, kaempferol; LL, linearolactone; N/A, not applicable; i.p., intraperitoneal route; Vh, vehicle (1× PBS, phosphate-buffered saline); ALA, amoebic liver abscess.

## Data Availability

All the data generated or analyzed during this study are included in this published article (as well as in the Appendix A). The raw data are available from the corresponding author upon reasonable request.

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
