# Peer review of "Toxicological Evaluation of Kaempferol and Linearolactone as Treatments for Amoebic Liver Abscess Development in Mesocricetus auratus"

_ijms, 2024, doi:10.3390/ijms251910633_

Round 1

Reviewer 1 Report

Comments and Suggestions for Authors

The study evaluated the toxicological effects of intraperitoneal administration of kaempferol and linearolacton against liver abscess in an animal model.

Introduction.

1) The authors already provide an indication of the advantages of their work compared to previous studies; this must be expanded to give a clear signal of the gaps in the literature that would be filled by publication of this manuscript.

2) The use of the animal model employed is not justified well and the relevant passage must be written better.

Methodology.

1) Animals. Please explain the methodology for allocation of animals to groups.

2) Animals. Please include a table with a summary of the protocol employed and the groups set during the animal phase of this study.

3) Control chemicals. Please describe all control chemicals and consumables used in this study.

4) Control procedures. Please explain in detail all control procedures that you followed during this work.

5) Analysis. How many repeat testings were performed? How did you treat the data in relation to analysis?

6) Analysis. I noticed the use of mean ± sd. Did the data have a normal distribution? This is not mentioned anywhere.

7) Did you carry out a quantitative analysis of the histological findings, by setting various lesion-related criteria and assigning a score to each (say, 1-4) according to severity? If not, can you please include such an analysis?

Results and Discussion.

I general, I do not favour a joint section for Results and Discussion. The authors should consider about separating them.

1) The section is unusually long and difficult to read. Please introduce further sub-sections and sub-sub-sections to make the flow of reading easier.

2) Please explain the clinical consequences of this study.

References.

These are OK.

Conclusions.

1) The conclusions are not fully in line with the findings. I understand that the authors wish to make some extrapolation, but please tone down in the revised version.

Language.

The manuscript requires extensive corrections in English language.

Overall.

Extensive modifications as indicated above and reevaluation for final recommendation.

Comments on the Quality of English Language

Language.

The manuscript requires extensive corrections in English language.

Reviewer 2 Report

Comments and Suggestions for Authors

The manuscript entitled “Toxicological study of kaempferol and linearolactone as treatments for amoebic liver abscess development in Mesocricetusauratus” by Luis Varela-Rodríguez et al. explored the toxicological and effectiveness of kaempferol (KP) and linearolactone (LL) for amoebic liver abscess (ALA) treatment in vitro and in vivo. The experiments result demonstrated that LL may have a better therapeutic effect while KP possess a better safety profile. Authors further use bioinformatics tool to provide a prediction of drug properties of these two compounds. My major concerns are attached.

1.    In the introduction and abstract, author mentioned that KP and LL combined with MTZ could be an alternative approach for ALA treatment. However, in the manuscript, there is no therapeutic evaluation of combination therapy and mono therapy. To support this statement, more experiment studies should be included. Besides, in the introduction part, the mechanism of action of all three drugs should be summarized to demonstrate why KP and LL could be used combine with MTZ or alone.

2.    Line 89, author used CHO-K1 and BEAS-2B for cytotoxicity test. Is there any particular reason to use these two cell lines?

3.    Line 98, please rephrase this sentence “These data suggest that the effect of both active principles is possibly due to the characteristic phenotype of the cells.” As it is not clear what this mean.

4.    Figure 1, author tested concentrations of KP and LL from 100 to 500 ug/mL and stated that the concentration induced 50% Viability of cells is IC50. However, IC50 stands for half-maximal inhibitory concentration, where the maximal inhibitory effect of different compounds varies. In this study, it is clear that the inhibitory effect reaches plateau with 400 ug/mL hand higher concentration as the viability does not change significantly with higher concentration treatment. Thus the potently of these two compounds are different. As a result, the IC50 tested in the study might be overestimated. To get the correct IC50, more concentration with serial dilution is needed.

5.    Table 1, how LD50 is calculated?

6.    In animal studies, the dose selection need to be further justified. Previous in vitro studies reveals that MTZ is at least 10 times more potent than LL and KP, why in the in vivo studies the dose are similar (5mg/kg and 10mg/kg)? Furthermore, the therapeutic effect seems similar among MTZ, KP and LL groups. How to explain the in vitro and in vivo result inconsistency?

7.    Table S1, in the treatments parameters, LL seems have a better effect. The hypothesis should be discussed in the Result.

8.    Line 583, author mentioned “Results predicting biological activity in humans indicate that KP may act as a kinase inhibitor and nuclear receptor ligand, while LL may act as a ligand of G protein-coupled receptors (GPCRs) or nuclear receptor ligand and modulate ion channels.” Besides the in silicon method, is there any publications found KP and LL or similar compounds have the hypothesized MOA?

9.    Please consider moving some of the supplemental figures into the manuscript as these supplemental figures are heavily mentioned in the context and are essential to support the key conclusion of this manuscript.

Comments on the Quality of English Language

Extensive editing of English language required. 

Round 2

Reviewer 2 Report

Comments and Suggestions for Authors

All of my concerns has been addressed. Thanks